# Complete Chloroplast Genomes of *Chlorophytum comosum* and *Chlorophytum gallabatense*: Genome Structures, Comparative and Phylogenetic Analysis

**DOI:** 10.3390/plants9030296

**Published:** 2020-03-01

**Authors:** Jacinta N. Munyao, Xiang Dong, Jia-Xin Yang, Elijah M. Mbandi, Vincent O. Wanga, Millicent A. Oulo, Josphat K. Saina, Paul M. Musili, Guang-Wan Hu

**Affiliations:** 1CAS key Laboratory of Plant Germplasm Enhancement and Specialty Agriculture, Wuhan Botanical Garden, Chinese Academy of Sciences, Wuhan 430074, China; jacintandunge.jn@gmail.com (J.N.M.); directx0831@163.com (X.D.); yangjxgz@163.com (J.-X.Y.); mkala@wbgcas.cn (E.M.M.); vincentokelo@gmail.com (V.O.W.); millicentoulo@gmail.com (M.A.O.); jksaina@wbgcas.cn (J.K.S.); 2Sino-Africa Joint Research Center, Chinese Academy of Sciences, Wuhan 430074, China; 3University of Chinese Academy of Sciences, Beijing 100049, China; 4East Africa Herbarium, National Museums of Kenya, P.O. Box 45166 00100 Nairobi, Kenya; pmutuku@museums.or.ke

**Keywords:** *C. comosum*, *C. gallabatense*, chloroplast genome, phylogenetic analysis

## Abstract

The genus *Chlorophytum* includes many economically important species well-known for medicinal, ornamental, and horticultural values. However, to date, few molecular genomic resources have been reported for this genus. Therefore, there is limited knowledge of phylogenetic studies, and the available chloroplast (cp) genome of *Chlorophytum* (*C. rhizopendulum*) does not provide enough information on this genus. In this study, we present genomic resources for *C. comosum* and *C. gallabatense*, which had lengths of 154,248 and 154,154 base pairs (bp), respectively. They had a pair of inverted repeats (IRa and IRb) of 26,114 and 26,254 bp each in size, separating the large single-copy (LSC) region of 84,004 and 83,686 bp from the small single-copy (SSC) region of 18,016 and 17,960 bp in *C. comosum* and *C. gallabatense*, respectively. There were 112 distinct genes in each cp genome, which were comprised of 78 protein-coding genes, 30 tRNA genes, and four rRNA genes. The comparative analysis with five other selected species displayed a generally high level of sequence resemblance in structural organization, gene content, and arrangement. Additionally, the phylogenetic analysis confirmed the previous phylogeny and produced a phylogenetic tree with similar topology. It showed that the *Chlorophytum* species (*C*. *comosum*, *C*. *gallabatense* and *C. rhizopendulum*) were clustered together in the same clade with a closer relationship than other plants to the *Anthericum ramosum*. This research, therefore, presents valuable records for further molecular evolutionary and phylogenetic studies which help to fill the gap in genomic resources and resolve the taxonomic complexes of the genus.

## 1. Introduction

*Chlorophytum* Ker-Gawl. is a large and taxonomically complex genus [1], distributed in the Old World tropics, particularly, in Africa (possibly the center of origin), Australia, Asia, Madagascar, and India [2,3], that comprises about 215 species, six subspecies, and 14 varieties [4,5] in the subfamily Agavoideae in the Asparagaceae family [6]. They are perennial rhizomatous herbs, commonly known as spider plants, based on the mosaic form of *C. comosum* leaves [7]. The genus is well-known for its medicinal characteristics [8]. Therefore, it has attracted the attention of scientists due to pharmacologically relevant steroidal and triterpenoidal saponin contents in their roots [9]. Other pharmacological activities such as aphrodisiac, anticancer, larvicidal, and immunomodulatory were also mentioned by Deore et al., 2008, and Khanam et al., 2013, [10,11]. *C. comosum* is used to treat burns, bronchitis, fractures [12], and also has antitumor activity [13]. There are several well-documented studies on phytochemistry and the medicinal properties of different species in this genus [14]. The leaves of *C. borivilianum* and *C. breviscapum* are used as a vegetable [15]. Additionally, some of the species are cultivated for their ornamental values such as *C. comosum* and *C. filipendulum* subsp. *amaniense,* and are also effective in controlling soil erosion [1].

The increased economic values of the *Chlorophytum* species have contributed to unregulated harvesting and habitat degradation. In addition, poor germination and regeneration rates have resulted in the scarcity of these plants. This has culminated in some species being listed as vulnerable, threatened, or critically endangered. *C. rhizopendulum* is one of the critically endangered species according to the International Union for Conservation of Nature (IUCN) Red List (www.iucnredlist.org). Moreover, this genus is experiencing difficulties in classification, and different taxonomists have misidentified the species when using morphological features [16]. *C. comosum* and *C. gallabatense* were reported to have these delimitation problems [17]. This problem of species delimitation has not been solved, and it is highly desirable to find other methods to authenticate the species of *Chlorophytum.* Phylogenetic analysis based on the shared protein-coding genes or complete (cp) genomes could solve these problems; however, the *Chlorophytum’s* complete (cp) genomic data is very insufficient. To date, *Chlorophytum* genomic data is underrepresented in literature, only one complete chloroplast genome (*C. rhizopendulum*) has been published in this genus [18], regardless of the chloroplast being illustrated as important for plant genetic molecular studies [19]. Consequently, this limits further phylogenetic analysis of this genus.

Chloroplasts are important organelles in plants, where carbon fixation, photosynthesis, and pigment synthesis are carried out [20,21], and therefore they are the main plastid for green plants [22]. The cp is an independent genome (chloroplast DNA) with a preserved, circular structure and a low molecular weight [23]. It has a typical quadripartite structure, consisting of a pair of inverted repeat regions (IRs) of equal size, separating a large single copy (LSC) region from a small single copy (SSC) region [24]. The plastid size ranges from 107 kb to 218 kb [25,26]. It contains approximately 110 to 130 genes encoding ribosomal RNA (rRNA) genes, protein-coding genes (PCGs), and transfer RNAs genes (tRNAs) [27]. Chloroplast genomes are easier and efficient to use for evolutionary studies because of their self-replication and slow evolutionary process that is relatively independent [28,29]. Recently, complete (cp) genome sequencing and assembly have improved, becoming less costly and simpler [20,30], as the next-generation sequence technology has evolved rapidly [31]. Studying the structure and sequence information of the cp genome gives researchers very important information for species conservation, construction of phylogeny [28,32], understanding genomic evolution, and solving relationship issues at various taxonomic levels [28,33]. In addition, it provides information on the genetic structure, present genes, gene order, and nucleotide alignment according to Gao et al., 2009 [30]. This is possible because the chloroplast genome is basic, stable, and conservative [34].

Currently, there is little data available on the genetic structures of *C. comosum* and *C. gallabatense,* especially on their detailed phylogenetic placement or chloroplast genomes. Only one nucleotide sequence of *Chlorophytum* (*C. rhizopendulum*) has been published (GenBank no. KX931454). In this study, we sequenced the complete cp genome sequences of *C. comosum* and *C. gallabatense* for the first time. We aimed to elucidate the structural features in the cp genomes, codon usage, RNA editing, simple sequence repeat (SSR), highly divergent regions, and phylogenetic analyses of *C. comosum* and *C. gallabatense,* as well as compare them with five Asparagales species (four Asparagaceae and one Amaryllidaceae species). Our findings provide a basic genetic tool for species identification, evolution, genetic engineering, population genetics, and phylogenetic studies of the species in genus *Chlorophytum*. Moreover, our results help to fill the gap in genomic resources and resolve the taxonomic complexes of the genus.

## 2. Results and Discussion

### 2.1. C. comosum and C. gallabatense Chloroplast Genome Features

The cp genomes of *C. comosum* and *C. gallabatense* are 154,248 bp and 154,154 bp in size, respectively. The two sequences contained four regions; the LSC region of 84,004 bp and 83,686 bp length, the SSC regions of 18,016 bp and 17,960 bp length, and a pair of inverted repeat regions (IRa and IRb) of 26,114 bp and 26,254 bp each in length, in *C. comosum* and *C. gallabatense*, respectively (Figure 1 and Table 1). The overall guanine-cytosine (GC) ratio was 37.3% in the two cp genomes (Table 1). Both *C. comosum* and *C. gallabatense* cp genome features were similar to other reported species in the Asparagaceae family with respect to gene content, order, and orientation [35].

We detected a total of 112 unique genes in *C. comosum* and *C. gallabatense* cp genomes, which included 78 PCGs, four rRNA, and 30 tRNA genes (Table 1). Amongst them, 19 genes are replicated in the IR regions, comprising seven PCGs (*ndhB*, *ycf1*, *ycf2*, *rpl2*, *rps7*, *rpl23*, and *rps12*), four rRNA genes (*rrn23*, *rrn4.5*, *rrn5*, and *rrn16*), and eight tRNA genes (*trnA*-*UGC*, *trnH*-*GUG*, *trnI*-*GAU*, *trnI*-*CAU*, *trnL*-*CAA*, *trnN*-*GUU*, *trnV*-*GAC*, and *trnR*-*ACG*) (Table 2). There are fifteen intron-containing genes, of which thirteen have one intron (*ndhA*, *atpF*, *ndhB*, *rpoC1*, *rpl2*, *rps16*, *rps12*, *trnV*-*UAC*, *trnl*-*GAU*, *trnL*-*UAA*, *trnK*-*UUU*, *trnG*-*UCC*, and *trnA*-*UGC*), whereas two genes (*ycf3* and *clpP*) contained two introns. In total, nine of these genes are PCGs and six are tRNA genes (Table 2). This result is similar with previous report by Sheng et al., 2017 [35].

The *rps12* gene is unequally distributed (trans-splicing gene), with the 5′ exon positioned in the LCS part, and the 3′ exons are positioned in the IRa/IRb regions. A similar observation in cp genomes of other species has been previously documented [36,37]. The presence of introns in the plant plastomes can lead to this expression of exogenous genes at precise locations and times [38]. Moreover, two pseudogenes (*rps2* and *infA*) were positioned in the LCS region. They are known as pseudogenes due to the reverse repetitive nature of the IR regions, and therefore not wholly duplicated, losing the capacity to encode whole proteins [39]. There were no significant rearrangements in gene order or inversions observed in comparisons of the two cp genomes of *Chlorophytum* with those of other selected genome sequences. Gene loss has been reported in some of the Asparagales species [40,41]. McKain et al., 2016 [18], confirmed that *rps19* was either pseudogenized in various positions or absent. Comparative analysis of seven species of Asparagales showed that the *rps19* gene was pseudogenized in all the species apart from the species of *Chlorophytum*, where *rps19* was missing as reported by McKain et al., 2016 [18].

### 2.2. Codon Usage Analysis

Codons are useful for the transmission of genetic information used in the evolution of genomes because they connect proteins and nucleic acids [42]. Codon usage is the use of similar codons with which an organism encodes the amino acids in the genes of their proteins [43]. This codon usage, however, has been shown to vary across different species [44] due to different factors such as codon hydrophilicity, tRNA abundance, gene length, gene expression rate, and base group composition [45]. Codon usage bias signifies that each gene of a species has its own preferred amino acid codon [46]. In the two species, the similarity was found in the codon usage and amino acid frequencies. The PCGs contained 20,479 and 20,453 codons in *C. comosum* and *C. gallabatense* cp genomes, respectively (Appendix A). A comparison was made with other selected species of Asparagales, and codon encoded the genes ranging from 19,986 (*Asparagus officinalis*) to 20,532 (*A. ramosum*), as shown in (Appendix A). In addition, leucine with a percentage of 10.41% (2,132) and 10.43% (2,133), was the most frequent amino acid in the cp genomes, followed by isoleucine with 8.77% (1797) and 8.75% (1789) in *C. comosum* and *C. gallabatense,* respectively. Cysteine, however, encoded the least frequently used amino acids with only 1.11% (228) and 1.12% (230) in *C. comosum* and *C. gallabatense,* respectively (Figure 2 and Appendix A). Cysteine are found less frequently due to their high sensitivity to changes in physiological and environmental conditions [47]. This was also similar to the rest of the Asparagaceae family (Appendix A).

If the relative value of synonymous codon usage (RSCU) is equal to one, the codon usage is not preferred (not biased), but highly preferred if the value is greater than one [36], indicating that the codon is used more often than expected [48] and less preferred with values of less than one. Almost all of the amino acid codons in *C. comosum* and *C. gallabatense,* cp genomes have preferences, which is due to the amino acid activity preventing error all through the transcription process, other than tryptophan (UGG) and methionine (AUG), which did not have a bias (RSCU values equal to one) (Appendix A). The results, moreover, showed that all preferred synonymous codons (RSCU > 1) have an A or U position at the third codon, except for UUG (Appendix A). In the cp genome, natural selection and direction of variation play a very significant role in influencing codon usage bias [49]. This analysis provides the required amino acids for protein biosynthesis in the cp genomes of *C. comosum* and *C. gallabatense*. These results are similar to cp genome reports of other terrestrial plants [50,51].

### 2.3. RNA Editing Sites

RNA editing is a common molecular process in the cp genomes of plants [52]. It has different functions that can change a transcribed RNA’s protein-coding sequence by altering, deleting, or inserting nucleotides during transcription [53]. The predictive plant RNA editor (PREP) software [54], which uses 35 genes as references for possible RNA editing sites by comparing the predicted genes to homologous protein genes from other plants, identified editing sites in 26 PCGs studied (Appendix A). As a result, in *C. comosum* and *C. gallabatense* a total of 64 and 66 potential RNA editing sites were predicted, respectively. The *ndhB* gene had the highest number (12 sites) in the two cp genomes of RNA editing sites followed by *ndhF*, *rpoB*, and *matK*, which had more than five editing sites. The gene *rpoC2*, *atpF*, *rpoC1*, *rps14*, *ycf3*, *accD*, *psbF*, *rpl20*, *clpP*, *petB*, *rpoA*, *rps8*, *rpl2*, *ccsA*, *ndhG*, and *ndhA* had editing sites more than one. The remaining five genes (*rps16*, *atpA*, *atpI*, *atpB*, and *petG*) did not have RNA predicting sites. Additionally, variations in amino acid as a result of RNA editing occurred most frequently in serine to leucine which reflects a conversion into a hydrophobic amino acid from a hydrophilic, while arginine to cysteine occurred less frequently (Appendix A). These general features of advanced RNA editing in the plants cp genomes have also been documented by previous studies [55,56] which found that important RNA editing sites direct amino acid to alter from polar to non-polar in order to form a fundamental organization of protein to enhance protein hydrophobicity [57].

### 2.4. Analysis of SSRs

Microsatellites or simple sequence repeats (SSRs) are tandemly repetitive DNA sequences, comprising of one to six (mono-, di-, tri-, tetra-, penta-, and hexa-) repeat nucleotide units. Also known as microsatellites, SSRs are good molecular markers often used in plant species phylogenetics, identification and population genetic studies [58], as they are highly reliable, reproductive, and highly polymorphic [37,44]. They are highly spread in the PCGs, introns, and intergenic regions. We detected a total of 111 SSRs in *C. comosum* and 90 SSRs in *C. gallabatense* (Figure 3 and Table 3). Likened to other Asparagales species, *A. ramosum* had the largest number of repeats with 73 SSRs, *C. rhizopendulum* with 71, *Allium victorialis* with 64 SSRs, *Asparagus officianalis* with 55 SSRs, and the least number of repeats in *Anemarrhena asphodeloides* with 49 SSRs. *C. comosum* was comprised of 86 mononucleotide repeats, 14 dinucleotide repeats, two trinucleotide repeats, and nine tetranucleotide repeats, and *C. gallabatense* included 68 repeats for mononucleotides, 12 repeats for dinucleotides, one repeat for trinucleotides, eight repeats for tetranucleotides, and one repeat for pentanucleotides. Ten hexanucleotides were only present in *A. ramosum* cp genome (Figure 4 and Appendix A). The mononucleotide SSRs had the richest content in all the species, followed by dinucleotides and tetranucleotide. In the cp genomes, short polyadenine (poly A/T) repeats are the most frequent SSRs as compared with guanine (G) or cytosine (C) repeats [59]. In this research, the A/T mononucleotide repeats were the most abundant type in all the cp genomes. This result supports another study where the A/T repeats were abundant [60]. Genetic diversity is reflected in the regions with high mutation rates [61]. The identified loci, in this study, could be good molecular markers, useful in the study of population genetics of *C. comosum* and *C. gallabatense* and phylogenetic study in the future.

### 2.5. IR Expansion and Contraction

Variations in the size of the cp genome are primarily due to the contraction and expansion of the border regions [62]. Such modifications have long-term effects on the size of cp genomes. According to Zhang et al., 2017 [63], IR regions are possible distinctive features in most of the angiosperms. The arrangement of genes in the junctions of cp genomes differs from one species to another [64]. Most of the cp genomes tend to be rearranged due to the loss of the IR regions [65], although, Chumley et al., 2006 [25], discovered that the *pelargonium* x *hortorum* genome was rearranged even without the loss of the IR regions. A comparison was made among Asparagales species on the border regions and the adjacent genes and some remarkable variations were discovered (Figure 5). Similar to other typical cp genomes, these genomes showed clear variances at the junctions, but the general gene structures, contents, and orientations were the same. The *rps19* gene was located at IRb, 37 bp, 49 bp, 53 bp, and 82 bp away from the junction of LSC/IRb in *A. ramosum*, *A. asphodeloides*, *A. officinalis*, and *A. victorialis,* respectively; however, this gene was missing in all three *Chlorophytum* species proving that the common ancestry lost this gene independently. The *rpl2* gene was positioned at the IRb in all chloroplast genomes at variable extensions from the junction. Similarly, the *psbA* gene occurred in the LSC region in all the genome sequences 1 bp to 101 bp away from IRa/LSC boundary. The distance from the IRa/SSC junction to *ndhF* gene differed by 5 bp in *A. asphodeloides* to 70 bp in *A. ramosum*. Still, there was an overlap between *ycf1*/*ndhF* in *A. victorialis*, *A. officinalis*, *C. comosum*, *C. gallabatense,* and *C. rhizopendulum*. Interestingly, the *ndhF* gene overlapped the SSC and IRb junction with 2 bp in *C. rhizopendulum* and 31 bp in *A. victorialis*. The border region in the middle of IRa and SSC was positioned within the *ycf1* gene, and the *ψycf1* pseudogene situated at the IRa region ranging from 802 bp to 1174 bp in size. The length of the SSC regions ranged from 17,853 bp in *A. victorialis* to 18, 638 bp in *A. officinalis*. The *ψycf1* gene was located at the junction between SSC and IRb. This pattern of IR expansion and contraction of partial copies of non-coding genes is a common occurrence in most terrestrial species [66]. These outcomes can provide insight into the evolutionary processes of chloroplast genomes, as well as being a source of DNA barcodes.

### 2.6. Analysis of Nucleotide Diversity

To examine the divergence of sequence in the cp genomes, the nucleotide variability (Pi) value was calculated using the DnaSP v5.10 program. A comparison was made with the other randomly selected five Asparagales species on the two cp genomes of *Chlorophytum*. The Pi values of the seven Asparagales species ranged from 0 to 0.15897 (Figure 6). This indicates that the species in the order Asparagales could be undergoing rapid nucleotide substitution [36]. The IR regions, as predicted, had a much lower nucleotide variability than the SSC and LSC regions. Other nucleotide diversity analyses have also shown that the IR region has a lower value of nucleotide diversity than the SSC and LSC regions [67,68]. This shows that IRa/IRb regions are much more conserved as compared with the LSC and the SSC regions as a result of gene replication [69]. Six divergent regions revealed higher nucleotide diversity values (Pi values > 0.1). All these variable regions were found in the intergenic spacer. This indicated that the intergenic regions were more divergent than the coding regions. Four divergent regions are located in LSC (*psbK*/*psbI*, *trnS*/*trnG*, *psbM*/*trnD*, and *rps12*/*clpP*), one in SSC (*rps15*/*ycf1*), and one in IR (*trnV*/*rps12*). Some of these regions have been previously reported in other species cp genomes [37,67]. Such highly variable regions can provide potential molecular markers for the authentication of plants and assist in phylogenetic analysis studies in this genus [70].

### 2.7. Comparative Genome Analysis

Comparative genomic analysis and the available DNA sequences make it possible to have a comprehensive view of a genus [71]. The cp genome sequences of *C. comosum* and *C. gallabatense* were compared to those of *C. rhizopendulum*, *A. victorialis*, *A. asphodeloides*, *A. officinalis,* and *A. ramosum* using the mVISTA program (Figure 7) and MAUVE program (Figure 8). Among the cp genomes compared, there were no significant rearrangements observed, except for the slight variations in size and gene positioning (Figure 8). *A. asphodeloides* (156,917 bp) was the largest, followed by *A. officinalis* (156,699 bp), then *A. ramosum* (155,812 bp), then *C. comosum* (154,248 bp), then *C. gallabatense* (154,154 bp), then *A. victorialis* (154,074 bp), and finally the shortest was *C. rhizopendulum* (153,504 bp). The results revealed that the genomes with some level of divergence were greatly conserved. The non-coding regions were less conserved as compared with the coding regions. The LSC and the SSC regions were equally less conserved than the IR regions. In addition, it was also detected that the intergenic spacer regions had the highest divergent regions. Some of the high divergent regions included *psbI-trnG*-*UCC*, *matK*-*rps16*, *psbZ*-*rps14*, *rps16*-*psbI*, *atpH*-*atpI*, *trnC*-*GCA*-*petN*, and *trnS*-*GGA*- *trnL*-*UAA* (Figure 7). The comparative study showed the conservation of gene order, signifying the evolutionary conservation of these species [71]. These regions can be used as unique barcodes for DNA and also give phylogenetic information.

### 2.8. Phylogenetic Analysis

Due to the rapid development of the sequencing technologies, cp genomes, which are valuable genomic resources, have been used to a greater extent in the reconstruction of plant phylogenies and evolutionary relationships [72]. The phylogeny of Asparagales was reconstructed using whole chloroplast genome sequences of 48 species belonging to the family Asparagaceae (thirty-eight species), the family Amaryllidaceae (eight species), and Iridaceae (two species) as outgroup. All the 46 cp genomes were downloaded from GenBank (Appendix A), and the 48 cp genomes were aligned by MAFFT v7.308 [73]. Then, the phylogeny was reconstructed using the maximum likelihood (ML) approach to reveal the evolutionary position of *C. comosum* and *C. gallabatense* and the relationship with other Asparagales species. Almost all of the nodes had high bootstrap support values (BP). The result indicated that genus *Chlorophytum* is closely related to genus *Anthericum.* The *Chlorophytum* species (*C. comosum*, *C. gallabatense,* and *C. rhizopendulum*) were clustered together in the same clade with a bootstrap value of 100%. Bjora et al., 2008 [17], in their study, stated that *C. gallabatense* clusters together with *C. comosum*, *C. filipendulum,* and *C. macrophyllum*. These species had a closer relationship to the *Anthericum ramosum* which formed an independent clade (Figure 9). This shows the affinity relationship among the species. During the early classification system, genus *Chlorophytum* was classified to be under the family Anthericaceae together with *Anthericum*. After several revisionary works, the two genera were separated using morphological characters (stamen filament ornamentation) and some of the species of the *Anthericum* were transferred into genus *Chlorophytum* [74], hence, the close relationship in the phylogenetic tree. These findings show that information from the whole cp genome sequences can give more reliable phylogenetic outcomes, therefore, more species of *Chlorophytum* can be sequenced in order to have a more comprehensive phylogenetic tree in this genus and give insight into the Asparagaceae relationship. Our results confirm the position of the selected Asparagales species supporting the previous whole genome and multigene-based analyses [35,75,76] (Figure 9). Therefore, this study has provided an in-depth molecular analysis within the family Asparagaceae and can be used in the taxonomy of the family especially using the molecular markers upon further analysis.

## 3. Materials and Methods

### 3.1. DNA Isolation and Chloroplast Genome Sequencing

Young, fresh, and healthy leaf samples of *C. comosum* and *C. gallabatense* were collected from Kenya (*C. comosum* SAJIT-004032 and *C. gallabatense* SAJIT-006319) and dried in silica gel to preserve the DNA [77]. All the samples were taken to the Herbarium of Wuhan Botanical Garden (HIB) and stored at −80 °C until DNA extraction. The genomic DNA was isolated from 100 micrograms of leaves using a modified cetyltrimethylammonium bromide (CTAB) technique [78] and sequenced using the Illumina platform at Novogene Company (Beijing, China).

### 3.2. Genome Assembly and Annotation

The clean data obtained after filtering the low-quality data and adaptors were assembled using GetOrganelle version 1.6.2 software [79], and then, manually corrected. The assembled chloroplasts were used for gene annotation by the GeSeq online tool (https://chlorobox.mpimp-golm.mpg.de/geseq.Html) with default settings [80]. Manual verification was done using Sequin Viewer to relate the cp genomes of *Chlorophytum* species with those of Asparagaceae. Annotations of tRNAs were confirmed using the tRNAscan-SE [81]. The boundaries of the introns and the start and stop codons were manually corrected. Gene maps of complete cp genome sequences were drawn using OrganellarGenome DRAW software [82] (Figure 1). The annotated cp genomes of the *C. comosum* and *C. gallabatense* were submitted to the GenBank (GenBank numbers: *C. comosum*: MT076065 and *C. gallabatense*: MT036265).

### 3.3. Genome Structure and Comparison

The genome features of *C*. *comosum* and *C*. *gallabatense* were determined using MEGA7 [83] by comparing the five other available cp genomes of Asparagales which were downloaded from NCBI (Appendix A). The PCGs from the *Chlorophytum* species and the four Asparagales species were extracted manually and the Predictive RNA Editor for plants (PREP) suite online software was used to identify the RNA editing sites [54] where the cutoff value was set to 0.8. PREP server. The default settings used 35 genes to compare and predict the potential RNA editing sites.

For comparative analysis, the cp genomes of *C. comosum* and *C. gallabatense* with the five randomly selected Asparagales species was constructed using the mVISTA program [84]. Furthermore, in the rearrangement analysis, the cp genomes were aligned and constructed in MAUVE program [85]. The expansion and contraction of IRs were analyzed using the IRscope online program (https://irscope.shinyapps.io/irapp/), then manually modified. To detect the divergence hotspots, the complete chloroplast genomes of the selected seven Asparagales species were aligned by MAFFT v7.308 [73]. DnaSP v5.10 was used to calculate the nucleotide divergence values of the chloroplast genome sequence alignment of seven analyzed species using the sliding window method [86] with a window length of 600 bp and a 200 bp step size.

### 3.4. Codon Usage

The codon usage frequency in each of the seven Asparagales species was analyzed for all the PCGs using MEGA7 [83]. Synonymous codon usage and the relative synonymous codon usage (RSCU) was conducted to determine if the plastid genes were under selection.

### 3.5. Simple Sequence Repeats

Simple sequence repeats (SSRs) present in cp genomes of *C*. *comosum* and *C*. *gallabatense* were analyzed using the software MISA (MicroSAtellite) [87] with the settings of ten repeat units for mononucleotides, five for dinucleotides and trinucleotides, and three for tetranucleotides, pentanucleotides, and hexanucleotide repeats.

### 3.6. Phylogenetic Analysis

To confirm the phylogenetic positions of *C*. *comosum* and *C*. *gallabatense* within the family Asparagaceae, we downloaded forty-six cp genomes representing three families of Asparagales (Amaryllidaceae, Asparagaceae, and Iridaceae as outgroup) from the NCBI database (Appendix A) for analysis. Multiple sequence alignment of the forty-eight complete cp genome sequences was done using MAFFT v7.308 [73] with default parameters and the maximum likelihood (ML) tree with default settings and 1000 bootstrap replicates was reconstructed using the program IQ-TREE v6.10 with the best fit model TVM + I + G4. TreeDyn [88,89] was used to visualize and refine the tree.

## 4. Conclusions

*Chlorophytum* is one of the essential genera in the family Asparagaceae as it is used for its medicinal properties, however, its cp genomes are not well studied. In this study, we analyzed the complete cp genomes of *C. comosum* and *C. gallabatense* of the family Asparagaceae for the first time. These genomes provided a basic genetic tool for species identification within the genus. We compared the cp genomes of *C. comosum* and *C. gallabatense* with five randomly selected Asparagales species. The results showed that the gene size, content, and order were all similar. The loss of *rps19* gene in the cp genomes of *Chlorophytum,* displays a distinctive feature of evolution within precise characteristics. The codon usage, RNA editing, and microsatellites were determined. The *ndhB* and *ndhF* genes had the highest number of editing sites. These results provide useful insights regarding editing sites that were gained or lost during the evolution of the Angiosperm genomes. Additionally, such findings provide genetic information for the genus *Chlorophytum* for the creation of molecular markers and for research into genetic diversity. The contraction and expansion of the boundary regions showed the genome size variation in the selected Asparagales species. Nucleotide analysis revealed the highest nucleotide diversity was in *rps12/clpP* and *rps15/ycf1* regions, which are highly variable regions that can be used as potential markers in family Asparagaceae for species identification and phylogeny. Additionally, the phylogenetic analysis confirmed the previous phylogeny and produced a phylogenetic tree with similar topology. It showed that the *Chlorophytum* species (*C. comosum*, *C. gallabatense,* and *C. rhizopendulum*) were clustered together in the same clade with a closer relationship than other plants to the *A. ramosum*. However, more complete cp genomes on *Chlorophytum* species should be sequenced which would help solve the relationship within Asparagaceae. Generally, this study provides valuable genetic information of *Chlorophytum* which can aid in further phylogenetic studies, species identification, and evolutionary relationships between *Chlorophytum* and *Anthericum*.

## Figures and Tables

**Figure 1 plants-09-00296-f001:**
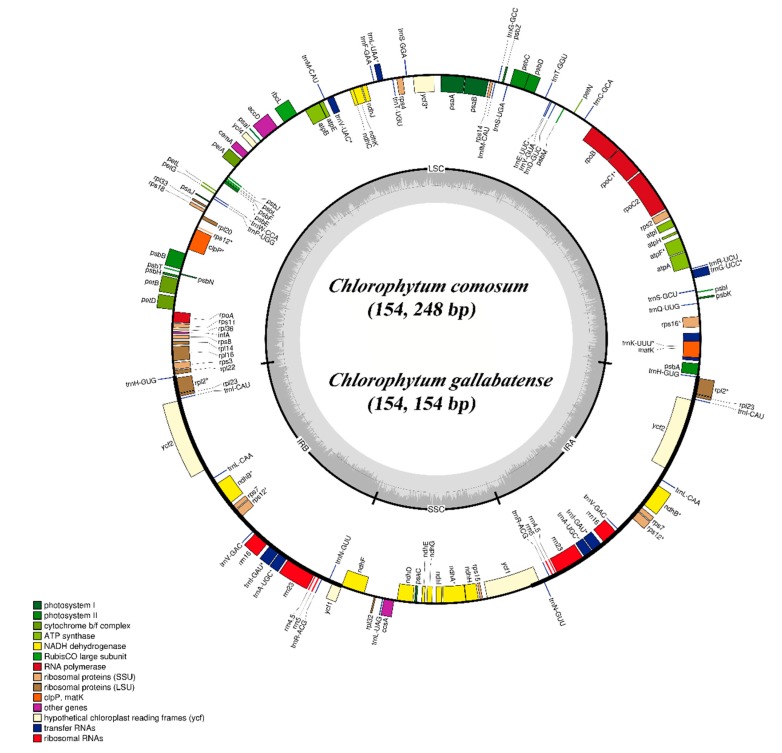
Gene map of the *C. comosum* and *C. gallabatense* complete cp genomes. Genes drawn within the circle are transcribed in the clockwise direction, and genes drawn out are transcribed in the counterclockwise direction. Genes are color-filled, basing on different functions. Inverted repeat (IR), small single-copy (SSC), and large single-copy (LSC) regions are indicated.

**Figure 2 plants-09-00296-f002:**
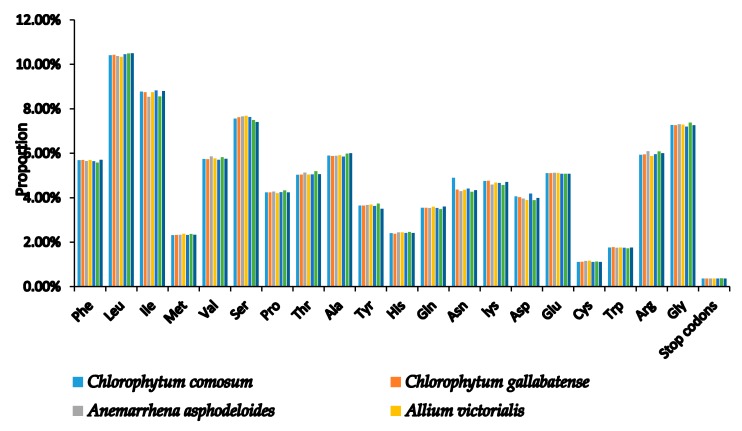
Amino acid proportion in *C. comosum* and *C. gallabatense* protein-coding sequences.

**Figure 3 plants-09-00296-f003:**
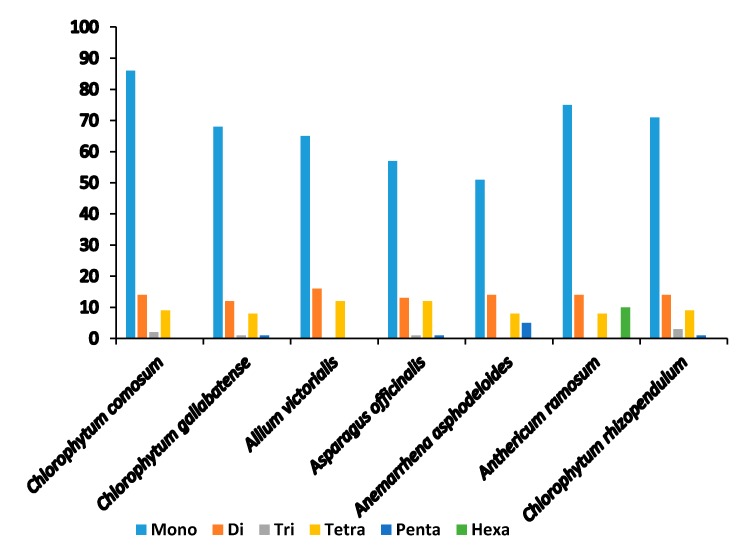
Number of different types of SSRs.

**Figure 4 plants-09-00296-f004:**
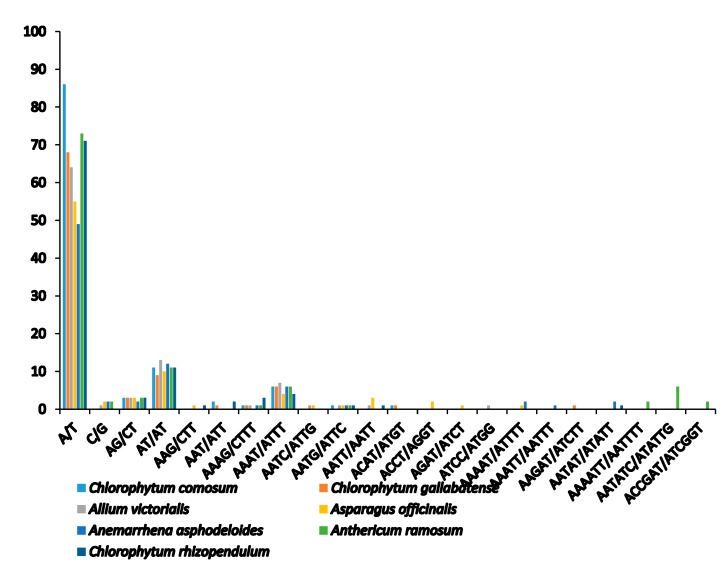
Number of different repeat units of SSRs.

**Figure 5 plants-09-00296-f005:**
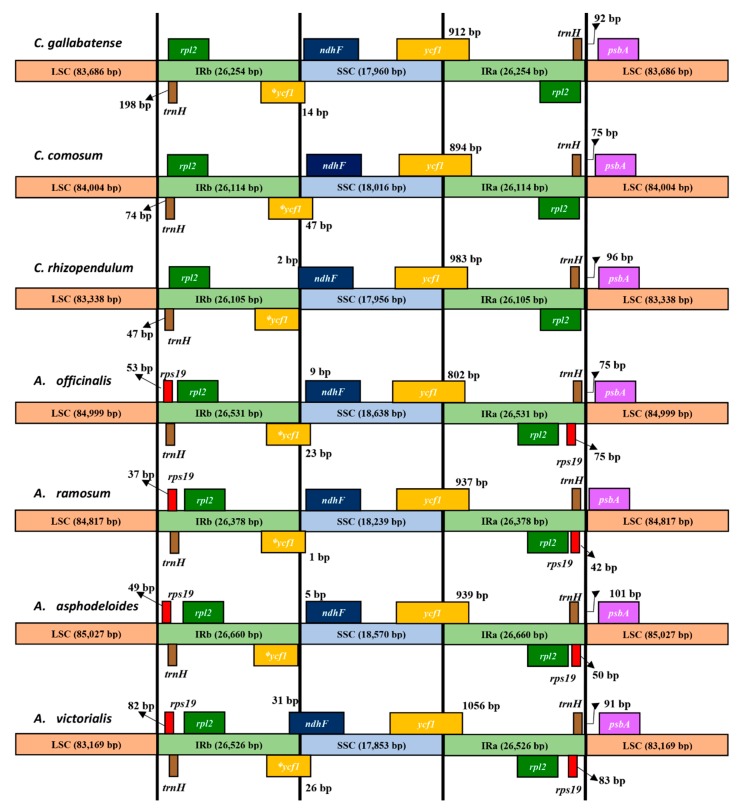
Comparison of the junctions of LSC, SSC, and IR regions among seven chloroplast genomes.

**Figure 6 plants-09-00296-f006:**
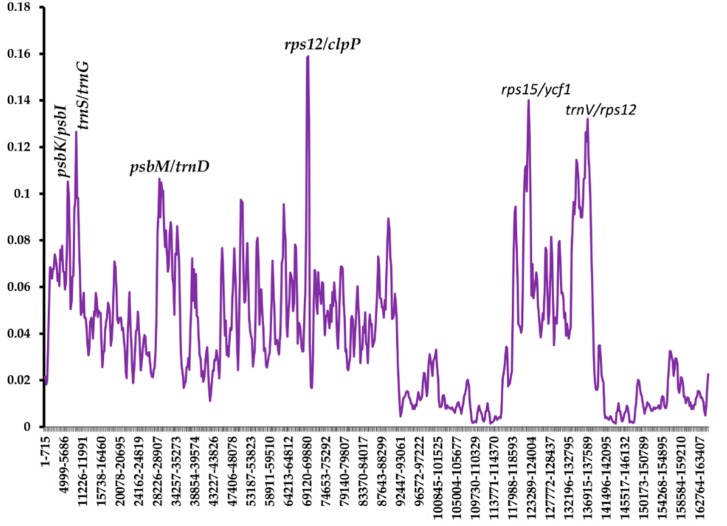
Nucleotide diversity of different regions of Asparagales chloroplast genomes.

**Figure 7 plants-09-00296-f007:**
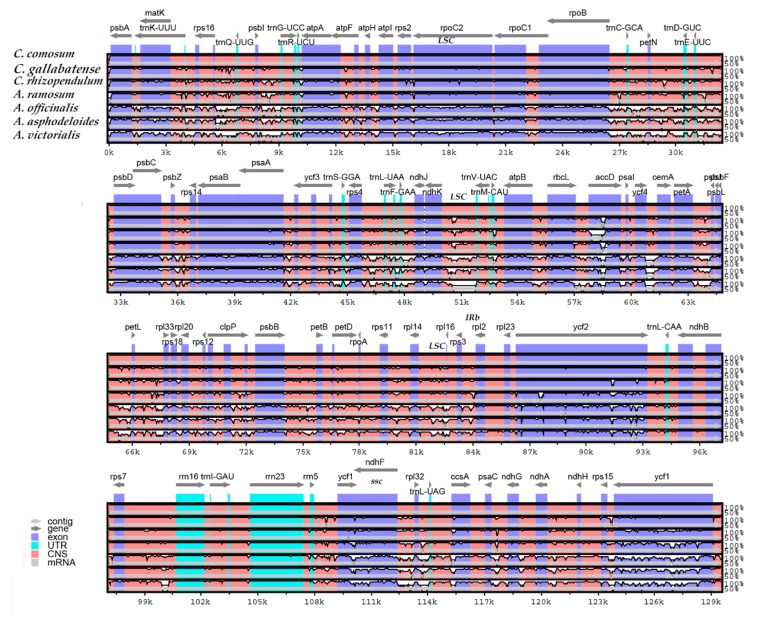
Comparison of seven cp genomes using mVISTA alignment program. Genome regions are color-coded as protein-coding, rRNA coding, tRNA coding, or conserved non-coding sequences. The vertical scale shows the percentage of identity, varying from 50% to 100%.

**Figure 8 plants-09-00296-f008:**
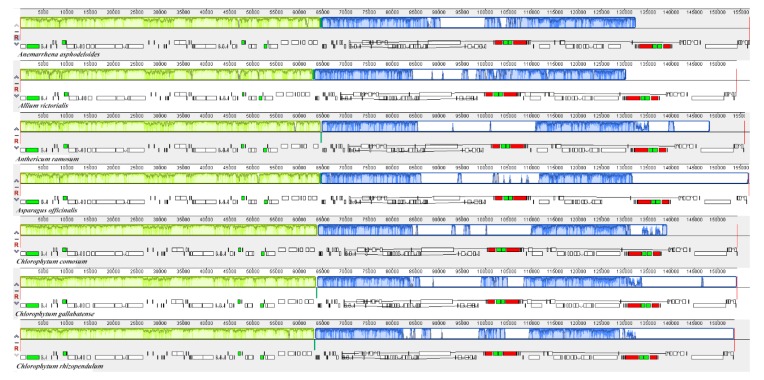
Comparison of the genome structure of seven Asparagales species using Mauve program. The DNA sequences above the line are presented in a clockwise direction, and those below the line in a counterclockwise direction.

**Figure 9 plants-09-00296-f009:**
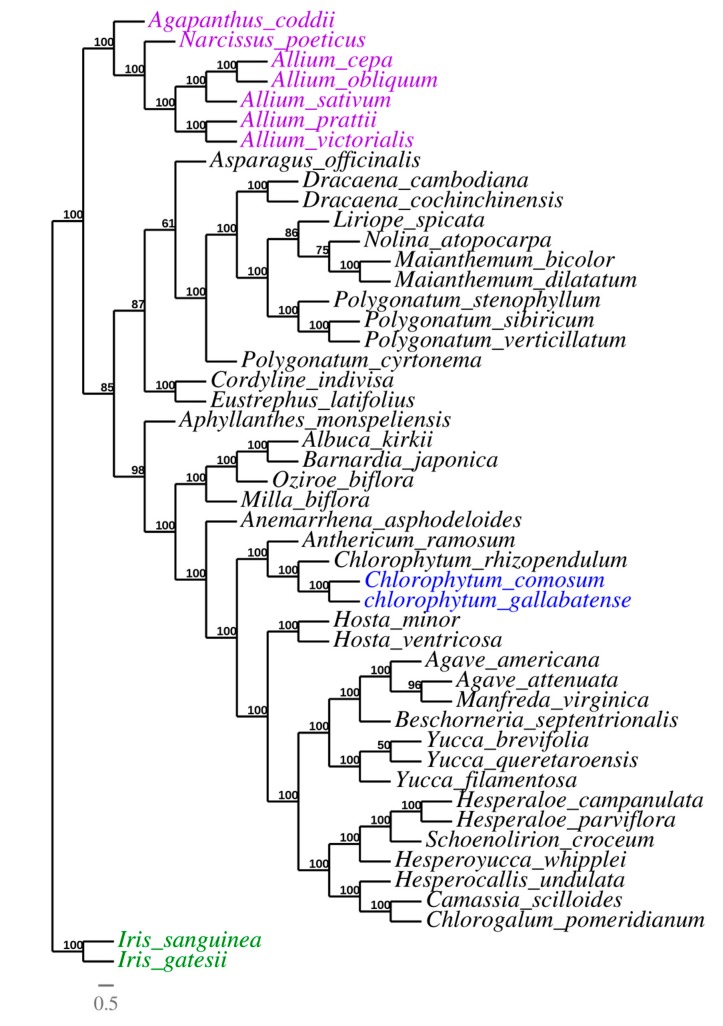
Phylogenetic trees of Asparagales species based on whole chloroplast genome sequences, with two species from family Iridaceae used as outgroup.

**Table 1 plants-09-00296-t001:** Features of the chloroplast genomes of *C. comosum* and *C. gallabatense* and the related Asparagaceae species.

Genome Features	*Chlorophytum comosum*	*Chlorophytum gallabatense*	*Allium victorialis*	*Asparagus officinalis*	*Anemarrhena asphodeloides*	*Anthericum ramosum*	*Chlorophytum rhizopendulum*
Size (bp)	154,248	154,154	154,074	156,699	156,917	155,812	153,504
LSC (bp)	84,004	83,686	83,169	84,999	85,027	84,817	93,446
IR (bp)	52,228	52,508	53,052	53,062	53,320	52,756	42,102
SSC (bp)	18,016	17,960	17,853	18,638	18,570	18,239	17,956
Total genes	112	112	116	112	112	114	112
PCGs	78	78	82	78	78	80	78
tRNA genes	30	30	30	30	30	30	30
rRNA genes	4	4	4	4	4	4	4
Adenine (A)	31.0%	31.0%	31.2%	30.9%	30.8%	31.0%	31.0%
Thymine (T)	31.7%	31.7%	31.8%	31.5%	31.4%	31.7%	31.7%
Guanine (G)	18.3%	18.3%	18.2%	18.5%	18.6%	18.3%	18.3%
Cytosine (C)	19.0%	19.0%	18.8%	19.1%	19.3%	19.0%	19.0%
GC content	37.3%	37.3%	37.0%	37.6%	37.8%	37.3%	37.3%

**Table 2 plants-09-00296-t002:** Genes present in *C. comosum* and *C. gallabatense* chloroplast genomes.

Category	Group of Genes	Name of Genes
Self-replication	Large subunit ribosomal proteins	*rpl2*^a,b^, *rpl14*, *rpl16*, *rpl20*, *rpl22*, *rpl23*^a^, *rpl32*, *rpl33*, *rpl36*
	Small subunit ribosomal proteins	*ψrps2*, *rps3*, *rps4*, *rps7*^a^, *rps8*, *rps11*, *rps12*^a,b^, *rps14*, *rps15*, *rps16*^b^, *rps18*
	DNA-dependent RNA polymerase	*rpoA*, *rpoB*, *rpoC1*^b^, *rpoC2*
	rRNA genes	*rrn4.5*, *rrn5*, *rrn16*, *rrn23*
	tRNA genes	*trnl-CAU*^a^, *trnL-CAA*^a^, *trnV-GAC*^a^, *trnl-GAU*^a,b^, *trnA-UGC*^a,b^, *trnR-ACG*^a^, *trnN-GUU*^a^, *trnL-UAG*, *trnP-UGG*, *trnW-CCA*, *trnQ-UUG*, *trnS-GCU*, *trnR-UCU*, *trnC-GCA*, *trnD-GUC*, *trnY-GUA*, *trnE-UUC*, *trnT-GGU*, *trnS-UGA*, *trnG-UCC*^b^, *trnG-GCC*, *trnfM-CAU*, *trnS-GGA*, *trnH-GUG*^a^, *trnT-UGU*, *trnL-UAA*^b^, *trnF-GAA*, *trnV-UAC*^b^, *trnM-CAU*, *trnK-UUU*^b^
photosynthesis	Photosystem I	*psaA*, *psaB*, *psaC*, *psal*, *psaJ*
	Photosystem II	*psbA*, *psbB*, *psbC*, *psbD*, *psbE*, *psbF*, *psbH*, *psbl*, *psbJ*, *psbK*, *psbL*, *psbM*, *psbN*, *psbT*, *psbZ*
	Cytochrome b/f complex	*petA*, *petB*, *petD*, *petG*, *petL*, *petN*
	ATP synthase	*atpA*, *atpB*, *atpE*, *atpF*^b^, *atpH*, *atpl*
	RuBisCO	*rbcL*
	Subunits of NADH-dehydrogenase	*ndhA*^b^, *ndhB*^a,b^, *ndhC*, *ndhD*, *ndhE*, *ndhF*, *ndhG*, *ndhH*, *ndhl*, *ndhJ*, *ndhK*
Other genes	Maturase	*MatK*
	Proteolysis	*clpP* ^c^
	Translation initiation factor	*ψinfA*
	Carbon metabolism	*cemA*
	Fatty acid synthesis	*accD*
	Cytochrome c synthesis gene	*ccsA*
unknown	Conserved open reading frames	*Ycf1*^a^, *ycf2*^a^, *ycf3*^c^, *ycf4*

^a^, two gene copies in IRs; ^b^, genes with one intron; ^c^, genes with two introns.

**Table 3 plants-09-00296-t003:** Numbers of simple sequence repeats (SSRs) in the *C. comosum* and *C. gallabatense* chloroplast genomes.

SSR Type	Repeat Unit	Amount		Ratio (%)	
		*C. comosum*	*C. gallabatense*	*C. comosum*	*C. gallabatense*
Mono	A/T	86	68	100%	100%
Di	AG/CT	3	3	21.4%	25%
	AT/AT	11	9	78.6%	75%
Tri	AAT/ATT	2	1	100%	100%
Tetra	AAAG/CTTT	1	1	11.1%	12.5%
	AAAT/ATTT	6	6	66.7%	75%
	AATG/ATTC	1	0	11.1%	0%
	ACAT/ATGT	1	1	11.1%	12.5%
Penta	AAGAT/ATCTT	0	1	0%	100%

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
