# Peer review of "Complete Chloroplast Genomes of *Chlorophytum comosum* and *Chlorophytum gallabatense*: Genome Structures, Comparative and Phylogenetic Analysis"

_plants, 2020, doi:10.3390/plants9030296_

Round 1
Reviewer 1 Report
The manuscript entitled “Complete chloroplast genome of Chlorophytum comosum and Chlorophytum gallabatense: genome structures, comparative and phylogenetic analysis” is a novel work providing the genomic resources and interesting data. The work is well designed and is worth publishing. However, the text needs to be further improved including English editing. For example, please put the abbreviated form of phrases only once when they are first written. In line 57, IUCN stands for what? In line 62 and 63, (cp) are written twice. Also, in line 74, “(rRNAs) ribosomal RNA genes” needs to be converted into “ribosomal RNA (rRNA) genes”.
In line 29 and main text, you cannot write as “C. rhizopendulum is the closest relative of Anthericum ramosum” without pairwise distance between three Chlorophytum species and Anthericum ramosum.
In line 140, what does ‘natural length’ and ‘mutagenesis expression rate’ mean?
In line 169, ‘The predictive plant RNA editor (PREP) software, of which 26 genes were analyzed, employed 35 genes in this analysis (Table S2)’. You analyzed the RNA editing only in the subset of genes but not the whole protein-coding genes?
In line 256~259, the unit ‘bp’ needs to be added within the parenthesis.
In line 314, please provide ID deposited in NCBI.
In Table 3, ‘Di’ needs to be written next to ‘AT/AT’.
In Figure 6, why did you calculate nucleotide diversity in the subset of regions but not the whole genome? How did you choose these regions? They are intergenic regions?
In Figure 8, how many bootstrapping did you conduct?
In Table 2 and main text, how many genes are pseudogenes? If infA and rps2 gene is pseudogene as you explained in line 124, please mark these genes with ψ in Table 2.
Author Response
On behalf of all authors.
Authors’ response
Thank you so much for the time spent to read this manuscript, and for the valuable comments and suggestions.
We have made the changes suggested by reviewer 1 as shown in the revised manuscript (Track changes mode).
Reviewer #1
Comments and suggestions for Authors
The manuscript entitled “Complete chloroplast genome of Chlorophytum comosum and Chlorophytum gallabatense: genome structures, comparative and phylogenetic analysis” is a novel work providing the genomic resources and interesting data. The work is well designed and is worth publishing. However, the text needs to be further improved including English editing. For example, please put the abbreviated form of phrases only once when they are first written.
In line 57, IUCN stands for what?
Thank you for your comment. We have included the full name of the initials- line 66
In line 62 and 63, (cp) are written twice.
We appreciate your careful comment. We have deleted “cp” in line 66 and 67.
In line 74, “(rRNAs) ribosomal RNA genes” needs to be converted into “ribosomal RNA (rRNA) genes”.
Agreed. We have revised and corrected- line 77
In line 29 and main text, you cannot write as “C. rhizopendulum is the closest relative of Anthericum ramosum” without pairwise distance between three Chlorophytum species and Anthericum ramosum.
Thank you so much, we completely agree. We have checked again and described the position of the species according to the phylogenetic tree.
In line 140, what does ‘natural length’ and ‘mutagenesis expression rate’ mean?
We appreciate so much the comment. We studied carefully the factors and realized that it should be “gene length” which is already included in the text and “gene expression rate”. Line 144
In line 169, ‘The predictive plant RNA editor (PREP) software, of which 26 genes were analyzed, employed 35 genes in this analysis (Table S2)’. You analyzed the RNA editing only in the subset of genes but not the whole protein-coding genes?
Thank you for the comment. All the protein-coding genes were analyzed. However, the predictive plant RNA editor (PREP) uses a total of 35 genes as reference for possible RNA editing sites, of which only 26 protein-coding genes were predicted to contain RNA editing sites. Line 176
In line 256~259, the unit ‘bp’ needs to be added within the parenthesis.
Agreed, we have revised accordingly. Line 272 and 273.
In line 314, please provide ID deposited in NCBI.
We apologize for this omission. We have submitted the annotated sequences to GenBank. However, we have received the accession number for one sequence Chlorophytum comosum: MT036265 in line 339 and 340, we have emailed the NCBI team and they promise to send the remaining accession number soon.
In Table 3, ‘Di’ needs to be written next to ‘AT/AT’.
We so much appreciate the comment. We revised accordingly.
In Figure 6, why did you calculate nucleotide diversity in the subset of regions but not the whole genome? How did you choose these regions? They are intergenic regions?
Thank you for the comment. We used the whole chloroplast genome to calculate the nucleotide variability. However, the six regions showed a higher nucleotide diversity values and all were intergenic regions. We have edited the text in materials and method part on how we performed the nucleotide diversity analysis. We did not use intergenic regions, introns or coding regions to calculate nucleotide diversity but could if you would like.
In Figure 8, how many bootstrapping did you conduct?
Thank you so much for the comment. The phylogenetic tree was constructed using ML analysis and 1000 bootstrap replicates (Default setting).
In Table 2 and main text, how many genes are pseudogenes? If infA and rps2 gene is pseudogene as you explained in line 124, please mark these genes with ψ in Table 2.
We so much appreciate the comment. There are two pseudogenes detected (rps2 and infA). Genes have been marked with “ψ”in table 2.
Thank you for your cautious and thoughtful comments. We have revised to improve English, results and the conclusion.

Reviewer 2 Report
Munyao, Dong et al. sequenced and assembled the complete chloroplast genomes of two species in the genus Chlorophytum, where the chloroplast genome sequence of another species was already available. They carried out various analyses of the chloroplast genome sequences, but most of the results shown in the manuscript are purely descriptive, lack clear insight for the readers of Plants, and mainly confirm or are similar to the results of previous studies. For example, they show the proportions of 20 amino acids for seven chloroplast genomes (Figure 2) and the conclusion is only that they are "similar to cp genome reports of other terrestrial plants.' The main weakness of the manuscript is that their conclusion in the abstract is not really supported by the results. While it says in line 28 the study "showed the close relationship between C. comosum and C. gallabatense, whereas C. rhizopendulum is the closest relative of Anthericum ramosum," the phylogenetic tree in Figure 8 shows that the three Chlorophytum species are clustered together.
Some minor comments:
The methods of phylogenetic analyses should be more clearly explained, including which sequences were used for alignment, the length of the alignment, etc. line 75: For what is it easier to use chloroplast? line 155: Are there any statistical analyses to support the "preferences" line 169: This sentence does not make sense to me.
Author Response
On behalf of all authors.
Authors’ response
Thank you so much for the time spent to read this manuscript, and for the valuable comments and suggestions.
We have made the changes suggested as shown in the revised manuscript (Track changes mode).
Reviewer #2
Comments and Suggestions for Authors
Munyao, Dong et al. sequenced and assembled the complete chloroplast genomes of two species in the genus Chlorophytum, where the chloroplast genome sequence of another species was already available. They carried out various analyses of the chloroplast genome sequences, but most of the results shown in the manuscript are purely descriptive, lack clear insight for the readers of Plants, and mainly confirm or are similar to the results of previous studies. For example, they show the proportions of 20 amino acids for seven chloroplast genomes (Figure 2) and the conclusion is only that they are "similar to cp genome reports of other terrestrial plants.' The main weakness of the manuscript is that their conclusion in the abstract is not really supported by the results. While it says in line 28 the study "showed the close relationship between C. comosum and C. gallabatense, whereas C. rhizopendulum is the closest relative of Anthericum ramosum," the phylogenetic tree in Figure 8 shows that the three Chlorophytum species are clustered together.
Some minor comments:
The methods of phylogenetic analyses should be more clearly explained, including which sequences were used for alignment, the length of the alignment, etc. line 75: For what is it easier to use chloroplast? line 155: Are there any statistical analyses to support the "preferences" line 169: This sentence does not make sense to me.
Thank you so much for your comments. We have revised the manuscript to improve our abstract, methods, results and the conclusion as shown in track changes mode
Line 75: For what is it easier to use chloroplast?
Thank you so much for your comment. In line 78, we have revised this part.
Line 155: Are there any statistical analyses to support the "preferences"
Thank you for the comment. Our statistical analyses are provided in “Table S1”.
Line 169: This sentence does not make sense to me.
Thank you for the comment. As explained in line 175, the predictive plant RNA editor (PREP) uses a total of 35 genes as reference for possible RNA editing sites, of which only 26 protein-coding genes were predicted to contain RNA editing sites.

Round 2
Reviewer 2 Report
The authors have corrected a major flaw in the abstract, which significantly improved the manuscript. I would suggest that they ask a native speaker to proofread the manuscript before publication.